# Diffusion Priors for Variational Likelihood Estimation and Image Denoising

**Jun Cheng, Shan Tan**[*]
School of Artificial Intelligence and Automation,
Huazhong University of Science and Technology
jcheng24@hust.edu.cn, shantan@hust.edu.cn

## Abstract

Real-world noise removal is crucial in low-level computer vision. Due to the remarkable generation capabilities of diffusion models, recent attention has shifted towards leveraging diffusion priors for image restoration tasks. However, existing diffusion priors-based methods either consider simple noise types or rely on approximate posterior estimation, limiting their effectiveness in addressing structured and signal-dependent noise commonly found in real-world images. In this paper, we build upon diffusion priors and propose adaptive likelihood estimation and MAP inference during the reverse diffusion process to tackle real-world noise. We introduce an independent, non-identically distributed likelihood combined with the noise precision (inverse variance) prior and dynamically infer the precision posterior using variational Bayes during the generation process. Meanwhile, we rectify the estimated noise variance through local Gaussian convolution. The final denoised image is obtained by propagating intermediate MAP solutions that balance the updated likelihood and diffusion prior. Additionally, we explore the local diffusion prior inherent in low-resolution diffusion models, enabling direct handling of high-resolution noisy images. Extensive experiments and analyses on diverse real-world datasets demonstrate the effectiveness of our method. Code is available at `https://github.com/HUST-Tan/DiffusionVI`.

## 1 Introduction

Real-world imaging modalities, such as photography and biomedical imaging, frequently encounter complex image noise that is both signal-dependent and spatially correlated [23, 35]. Removing such noise is critical for subsequent image analysis and understanding. Existing deep learning-based image denoising methods rely on either large amounts of paired images for supervised training [50, 48, 49, 25] or noisy images for self-supervised training [31, 23, 46, 33]. However, collecting massive amounts of data is expensive and time-consuming in real-world scenarios. Therefore, designing effective and data-efficient real-world denoising methods is of significant practical importance.

Incorporating image priors and the likelihood, and conducting the corresponding posterior inference (e.g., maximum-a-posteriori (MAP) estimation and mean estimation), is a classical and data-efficient approach to image restoration [26, 18]. Nowadays, deep generative models such as VAE, GAN, and Normalizing-flow have shown the capacity to capture and model complex image statistics, surpassing traditional analytical image priors [32]. Recent diffusion models have demonstrated state-of-the-art image generation capabilities [14, 37] and have been incorporated into various image restoration tasks as powerful image priors [19, 12, 5].

---

[*]Corresponding Author

38th Conference on Neural Information Processing Systems (NeurIPS 2024).

The essence of applying diffusion priors to address image denoising lies in accurately integrating the degraded image into the generation process of pre-trained diffusion models, underscoring the importance of the likelihood function. Unlike Gaussian white noise, real-world image noise is intricate and challenging to model precisely and analytically [17, 21]. Many existing methods based on diffusion priors solely account for $i.i.d.$ Gaussian noise [19, 5, 43, 10], making them ineffective for real-world noise removal. On the other hand, some approaches targeting complex and non-linear degradations employ *hard* data-consistency strategies and approximate posterior inference during the generation process [12, 45, 8], yielding unsatisfactory real-world denoising outcomes due to their coarse likelihood modeling. Although a structured and heteroscedastic Gaussian likelihood function can well approximate real-world noise, such a model is computationally expensive due to the large covariance matrix and also impractical due to the unknown noise variance.

To tackle these challenges, this paper integrates variational Bayes and presents adaptive likelihood estimation and MAP inference during the generation process of diffusion priors to handle real-world noise. We introduce an independent, non-identically distributed ($i.ni.d.$) likelihood combined with a precision prior to model real-world noise. Such a choice allows modeling the spatially variant feature of noise and meanwhile avoids modeling covariance, trading off the accuracy for practical feasibility. Based on variational Bayes, the $i.ni.d.$ precision posterior at each step in the reverse process is subsequently inferred, which adaptively refines the likelihood function and aligns with the real-world noise model. Additionally, we introduce local Gaussian convolution to rectify the estimated noise variance, compensating for the lack of spatial correlation in the $i.ni.d.$ likelihood function. By adaptively updating the likelihood at each reverse diffusion step, the final denoised result is achieved by progressively propagating the intermediate MAP solutions that strike the best balance between the noisy image and diffusion prior.

Furthermore, real-world images exhibit diverse resolutions, often differing from those of pre-trained diffusion models. Existing methods utilize patch-based [12] or resize-based [19] operations, which are laborious and may impact low-level details. We observe that diffusion models pre-trained with low-resolution (LR) images tend to yield local diffusion priors effective for image restoration, enabling the direct treatment of high-resolution (HR) noisy images. The main contributions of this paper are summarized as follows:

- We propose adaptive likelihood estimation and MAP inference based on diffusion priors and variational Bayes to address real-world complex noise.

- We explore the local prior exhibited by diffusion models pre-trained with LR images.

- Our method outperforms other unsurpervised denoising methods as well as diffusion priors-based methods on diverse real-world image denoising datasets.

## 2 Related Works

**Deep Learning-based Image Denoising**. By harnessing modern deep architectures and large-scale paired training datasets, supervised learning-based denoising methods such as VDN [48], Restormer [49], and GRL [25] have significantly enhanced in-distribution denoising performance. However, their reliance on extensive paired data poses challenges for real-world applications, prompting the exploration of self-supervised denoising approaches. These include Blind spot (BS)-based methods (e.g., SSDN [22], Noise2Self [3], Nei2Nei [16], and B2U [44]), resampling-based methods (e.g., Nr2N [29] and R2R [34]), and regularization-based methods (e.g., Noise2Score [20] and Stein [38]). However, these methods assume spatially independent or analytical noise, which deviates from the structured and complex real-world noise. Recent advancements, such as AP-BSN [23] and LG-BPN [46], have integrated Pixel-shuffle and masked convolution into BS networks to address real-world noise. Other approaches have leveraged disentangled representation learning [11, 31, 6]. Nonetheless, these methods often require large quantities of noisy images and lack data efficiency. Several single image-based deep learning methods (e.g., DIP [42], Self2Self [36], ZS-N2N [27], and ScoreDVI [7]) have been proposed, but their performance in real-world denoising scenarios remains suboptimal, underscoring the need for more effective approaches.

**Diffusion Priors for Image Restoration**. Diffusion models have exhibited remarkable image generation capabilities [14, 40, 37] and have been integrated into inverse problems as diffusion priors to address various image restoration tasks in an unsupervised manner [24]. Existing methods based

on diffusion priors can be broadly categorized into two aspects: linear inverse problem-solving based on exact degradation models (e.g., DDRM [19], DDNM [43], MCGDiff [5], and FPS [10]), and nonlinear inverse problem-solving based on approximate posterior inference (e.g., DPS [8], GDP [12], and DR2 [45]). In the former, these methods typically assume analytical noise, such as additive white Gaussian noise, overlooking real-world noise that is signal-dependent and spatially correlated. Given the difficulty in precisely modeling and estimating real-world noise, these methods often prove ineffective in handling such noise.

In the latter, *hard* data-consistency terms are introduced to replace accurate likelihood modeling, and approximate posterior inference is conducted during the inverse diffusion process to address complex degradations. However, due to the absence of explicit constraints from likelihood functions, these methods heavily rely on proper hyperparameters (e.g., guidance scales in GDP [12], step size in DPS [8], or downsampling factors in DR2 [45]), leading to significant reconstruction errors. Diverging from these methods, we refrain from specifying the accurate noise model but introduce the noise precision prior and dynamically estimate its posterior using variational Bayes in the reverse diffusion process, enabling adaptive estimation of likelihood functions and better posterior inference.

## 3 Methods

### 3.1 Preliminary

The Diffusion model is a class of generative models used to model the distribution $q(x_0)$. Its forward process is a Markov chain with fixed Gaussian transition and length $T$, which gradually corrupts the data $x_0$ by adding Gaussian noise according to a pre-defined variance schedule $\eta_1, \cdots, \eta_T$:

$$q(x_{1:T}|x_0) = \prod_{t=1}^{T} q(x_t|x_{t-1}), q(x_t|x_{t-1}) = \mathcal{N}(x_{t-1}; \sqrt{1 - \eta_t}x_{t-1}, \eta_t I) \tag{1}$$

A nice property of the forward process is that sample $x_t$ at any step $t$ can be obtained from $x_0$ in a closed-form manner:

$$q(x_t|x_0) = \mathcal{N}(x_t; \sqrt{\bar{a}_t}x_0, (1 - \bar{a}_t)I) \rightarrow x_t = \sqrt{\bar{a}_t}x_0 + \sqrt{1 - \bar{a}_t}\epsilon \tag{2}$$

where $a_t = 1 - \eta_t, \bar{a}_t = \prod_{s=1}^{t} a_s, \epsilon \sim \mathcal{N}(0, I)$.

In the reverse process, the diffusion model progressively recovers data from noise distribution $p(x_T)$, which is again a Markov chain with learned Gaussian transition:

$$p(x_{0:T}) = p(x_T) \prod_{t=1}^{T} p(x_{t-1}|x_t), p(x_{t-1}|x_t) = \mathcal{N}(\mu_\theta(x_t, t), \sigma_t^2 I) \tag{3}$$

where $\sigma_t^2$ is a constant relating to $\eta_t$ and can be pre-computed. And $\mu_\theta(x_t, t)$ is usually parameterized by a DNN $\epsilon_\theta(x_t, t)$:

$$\mu_\theta(x_t, t) = \frac{1}{\sqrt{a_t}} \left( x_t - \frac{\beta_t}{\sqrt{1 - \bar{a}_t}} \right) \epsilon_\theta(x_t, t) \tag{4}$$

### 3.2 Naive Image Denoising

Consider the image formation process $y_0 = f(x_0, n)$ in the real-world scenario, where $n$ is the raw noise, $f$ is the transformation function, $y_0 \in \mathbb{R}^N$ and $x_0 \in \mathbb{R}^N$ ($N$ is the total pixel number) are the observed noisy image and original clean image, respectively. Noise in $y_0$ generally exhibits signal-dependent and spatially-correlated characteristics, e.g., sRGB noise [23], due to the Poisson nature of photons and compound transformation in $f$. Suppose there are no non-linear parts in $f$, the image formation process can be simplified as $y_0 = x_0 + n_0(x_0)$ with $\text{corr}(n_0^i, n_0^j) > 0$, where $n_0$ is the image noise related to the signal $x_0$; $\text{corr}(i, j)$ represents the correlation coefficient between two neighboring elements $i$ and $j$.

Real-world image denoising task is to recover clean and high-quality $x_0$ from noisy $y_0$, which turns into solving the posterior $p(x_0|y_0)$ in Bayesian statistics. As the pre-trained diffusion model possesses superior image priors, it is natural to inject the observed information $y_0$ into its reverse diffusion

(or generation) process defined in Eq. (3) to achieve the conditional inference of $x_{0:T}$ given $y_0$, i.e., $p(x_{0:T}|y_0)$. Due to that $y_0$ and intermediate $x_t$ exhibit large distribution gaps (as they have different noise types and strength) and it is also difficult to directly model $p(y_0|x_t)$ [8], we hence follow [39, 45] and re-corrupt $y_0$ using Eq. (2) in each step $t$ to obtain $y_t$, which serves as intermediate conditions. Such choice results in the target conditional distribution:

$$p(x_{0:T}|y_0) \to p(x_T) \prod_{t=1}^{T} p(x_{t-1}|x_t, y_{t-1}) \propto p(x_T) \prod_{t=1}^{T} p(x_{t-1}|x_t) p(y_{t-1}|x_{t-1}) \quad (5)$$

where $y_t = \sqrt{\bar{a}_t} y_0 + \sqrt{1 - \bar{a}_t}\epsilon$, and $p(x_T|y_T) \approx p(x_T)$ as $x_T$ and $y_T$ are approximated independent normal distributions.

**Structured and heteroscedastic Gaussian likelihood model**. The right-hand side of Eq. (5) explicitly involves the prior $p(x_{t-1}|x_t)$ and likelihood $p(y_{t-1}|x_{t-1})$ at each step $t$. The prior $p(x_{t-1}|x_t)$ has been available from the pre-trained diffusion model in Eq. (3), necessitating the modeling of $p(y_{t-1}|x_{t-1})$. As discussed above, in principle we can assume that $y_0$ follows the structured and heteroscedastic Gaussian distribution $\mathcal{N}(x_0, \Sigma(x_0))$, where $\Sigma$ is the *non-diagonal* covariance matrix with variances related to signal $x_0$, which allows modeling the signal-dependent and spatially-correlated properties of real-world noise. As a result, we derive that

$$y_{t-1} = \sqrt{\overline{\alpha}_{t-1}}(x_0 + A\epsilon_2) + \sqrt{1 - \overline{\alpha}_{t-1}}\epsilon = x_{t-1} + \sqrt{\overline{\alpha}_{t-1}}A\epsilon_2, AA^T = \Sigma(x_0), \epsilon_2 \sim \mathcal{N}(0, I) \quad (6)$$

which indicates that $p(y_{t-1}|x_{t-1}) = \mathcal{N}(y_{t-1}; x_{t-1}, \Sigma(x_{t-1}))$ with $\Sigma(x_{t-1}) = \overline{\alpha}_{t-1}\Sigma(x_0)$. See detailed derivation in the Appendix A.1.

As the prior $p(x_{t-1}|x_t)$ in Eq. (3) and likelihood $p(y_{t-1}|x_{t-1})$ in Eq. (6) both have Gaussian forms, the posterior distribution in Eq. (5) is theoretically computable. Nevertheless, we note that the above formulation presents several practical challenges. First, specifying an accurate $\Sigma(x_0)$ for $y_0$ is *difficult*, which involves the estimation of noise variance and the noise correlation between neighboring pixels. These estimations are hard to achieve based on a single $y_0$ and are open research problems [17, 21]. In addition, the posterior inference with non-diagonal covariance matrix $\Sigma$ is both memory-demanding and computationally expensive. These challenges prevent the direct application of the structured heteroscedastic Gaussian likelihood.

## 3.3 Variational Denoising with Adaptive Likelihood Estimation

To deal with these difficulties, we consider $p(y_0|x_0) = \mathcal{N}(x_0, \text{diag}(\phi_0)^{-1})$, which has diagonal precision matrix $\text{diag}(\phi_0)$ (i.e., the inverse of the covariance matrix, $\phi_0 \in \mathbb{R}^N$). Such diagonal Gaussian likelihood allows modeling the spatially variant feature of real-world noise but ignores the noise correlation at this stage. Based on Eq. (6), $p(y_{t-1}|x_{t-1})$ then becomes:

$$p(y_{t-1}|x_{t-1}, \phi_{t-1}) = \mathcal{N}(y_{t-1}; x_{t-1}, \text{diag}(\phi_{t-1})^{-1}), \phi_{t-1} = \frac{\phi_0}{\overline{\alpha}_{t-1}} \quad (7)$$

**Hyperprior for precision** $\phi_t$. Instead of specifying an accurate $\phi_0$, which is again difficult, we introduce the independent Gamma hyperprior $p(\phi_0) = \prod_{i=1}^{N} \text{Gamma}(\phi_0^i; \alpha, \beta)$ for $\phi_0$ ($\alpha$ and $\beta$ are scalars), which serves as the *rough* precision prior for noise in $y_0$. Meanwhile, based on Eq. (7), it is straightforward that $\phi_{t-1}$ follows

$$p(\phi_{t-1}) = \prod_{i=1}^{N} \text{Gamma}(\phi_{t-1}^i; \alpha_{t-1}, \beta_{t-1}), \text{with } \alpha_{t-1} = \alpha, \beta_{t-1} = \beta\overline{\alpha}_{t-1} \quad (8)$$

Because $p(\phi_{t-1})$ merely provides initial gauss about the noise precision (also variance) at each step $t$, we then expect to find the corresponding precision posterior $p(\phi_{t-1}|x_{t-1}, y_{t-1})$, which is more accurate and allows a better likelihood function $p(y_{t-1}|x_{t-1})$. As posterior $\phi_{t-1}$ depends on $x_{t-1}$, we have to simuteniously infer them together, i.e., the following joint distribution:

$$p(x_{t-1}, \phi_{t-1}|x_t, y_{t-1}) = \frac{p(y_{t-1}|x_{t-1}, \phi_{t-1})^{\frac{1}{\gamma}} p(\phi_{t-1}) p(x_{t-1}|x_t)}{p(y_{t-1}|x_t)} \quad (9)$$

where $\gamma \leq 1$ is the temperature parameter, which is typically utilized in variational inference to scale the likelihood function [2].

**Variational inference of precision posterior**. As $p(x_{t-1}, \phi_{t-1}|x_t, y_{t-1})$ in Eq. (9) is a non-trivial distribution, we hence choose a trivial and factorized variational distribution $g(x_{t-1}, \phi_{t-1}) = g(x_{t-1})g(\phi_{t-1})$ to approximate the true posterior $p(x_{t-1}, \phi_{t-1}|x_t, y_{t-1})$, under the KL-divergence distance. Following the mean-field variational Bayes presented in [4], the optimal $g(x_{t-1})g(\phi_{t-1})$ can be solved by cycling through $x_{t-1}$ and $\phi_{t-1}$ and replacing each in turn with a revised estimate of $g(x_{t-1})$ and $g(\phi_{t-1})$. Specifically, we derive the following alternate update scheme for finding the optimal $g(\phi_{t-1})$:

**1. Update** $g(x_{t-1})$. Given $g(\phi_{t-1})$, the optimal $g^*(x_{t-1})$ is provided by

$$\log g^*(x_{t-1}) = \mathrm{E}_{\phi_{t-1}} \log p(y_{t-1}|x_{t-1}, \phi_{t-1})^{\frac{1}{\gamma}} p(x_{t-1}|x_t) p(\phi_{t-1}) \tag{10}$$

which corresponds to

$$g^*(x_{t-1}) = \mathcal{N}(x_{t-1}; \hat{\mu}_{t-1}, \hat{\sigma}_{t-1}^2) \tag{11}$$

with mean $\hat{\mu}_{t-1}$ and variance $\hat{\sigma}_{t-1}^2$ as

$$\hat{\mu}_{t-1} = \frac{\sigma_t^2 \mathrm{E}(\phi_{t-1}) \odot y_{t-1} + \mu_\theta(x_t, t)\gamma}{\mathrm{E}(\phi_{t-1})\sigma_t^2 + \gamma}, \hat{\sigma}_{t-1}^2 = \mathrm{diag}\left(\frac{\gamma\sigma_t^2}{\mathrm{E}(\phi_{t-1})\sigma_t^2 + \gamma}\right) \tag{12}$$

where $\odot$ denotes element-wise multiplication; $\mathrm{E}(\phi_{t-1})$ at step $T$ is initialized as a constant (which is robust to different initializations) and then is updated as $\mathrm{E}(\phi_{t-1}) = \hat{\alpha}_{t-1}/\hat{\beta}_{t-1}$ (see **Update 2**).

**2. Update** $g(\phi_{t-1})$. Similar to $g(x_{t-1})$, the optimal $g^*(\phi_{t-1})$ given $g(x_{t-1})$ is

$$g^*(\phi_{t-1}) = \prod_{i=1}^{N} \mathrm{Gamma}(\phi_{t-1}^i; \hat{\alpha}_{t-1}^i, \hat{\beta}_{t-1}^i) \tag{13}$$

with shape $\hat{\alpha}_{t-1}$ and rate $\hat{\beta}_{t-1}$ as

$$\hat{\alpha}_{t-1}^i = \alpha_{t-1} + \frac{1}{2\gamma}, \hat{\beta}_{t-1}^i = \beta_{t-1} + \frac{(y_{t-1}^i - \hat{\mu}_{t-1}^i)^2 + (\hat{\sigma}_{t-1}^2)^i}{2\gamma} \tag{14}$$

The detailed derivation of Eqs. (11) and (13) is given in the Appendix A.2.

**MAP estimation with updated likelihood**. During the updates, $\hat{\alpha}_{t-1}$ and $\hat{\beta}_{t-1}$ in $g(\phi_{t-1})$ are adptively updated and become signal-dependent as indicated by Eq. (14). Once the cycling converges, we can obtain the approximated posterior distribution $g(\phi_{t-1})$ and the updated likelihood $p(y_{t-1}|x_{t-1}) = \mathrm{E}_{\phi_{t-1} \sim g(\phi_{t-1})} p(y_{t-1}|x_{t-1}, \phi_{t-1})$ at step $t$. As a result, by maximizing the conditional distribution in Eq. (5) at step $t$, the optimal $x_{t-1}^*$ that balances the image prior and the observed $y_{t-1}$ can be obtained as follows:

$$\begin{aligned}
x_{t-1}^* &= \mathrm{argmax}\ \log p(y_{t-1}|x_{t-1}) + \log p(x_{t-1}|x_t) \\
&\approx \mathrm{argmax}\ \mathrm{E}_{\phi_{t-1}} \log p(y_{t-1}|x_{t-1}, \phi_{t-1}) + \log p(x_{t-1}|x_t) \\
&= \mathrm{argmax}\ -(x_{t-1} - y_{t-1})^2 \mathrm{E}(\phi_{t-1}) - \frac{(x_{t-1} - \mu_\theta(x_t, t))^2}{\sigma_t^2} \\
&= \hat{\pi}_{t-1} y_{t-1} + (1 - \hat{\pi}_{t-1})\mu_\theta(x_t, t), \mathrm{with}\ \hat{\pi}_{t-1} = \frac{\sigma_t^2}{\sigma_t^2 + 1/\mathrm{E}(\phi_{t-1})}
\end{aligned} \tag{15}$$

where $\mathrm{E}(\phi_{t-1}) = \hat{\alpha}_{t-1}/\hat{\beta}_{t-1}$ is the expectation of noise precision posterior (and hence $1/\mathrm{E}(\phi_{t-1}) = \hat{\beta}_{t-1}/\hat{\alpha}_{t-1}$ is the estimated noise variance at step $t$), and $\hat{\pi}_t \in [0, 1]$ suggests that the optimal $x_{t-1}^*$ is the convex combination of $y_{t-1}$ and $\mu_\theta(x_t, t)$. Note that, in the second row of Eq. (15), we utilize the following Jensen's Inequality

$$\log p(y_{t-1}|x_{t-1}) = \log \mathrm{E}_{\phi_{t-1} \sim g(\phi_{t-1})} p(y_{t-1}|x_{t-1}, \phi_{t-1}) \geq \mathrm{E}_{\phi_{t-1} \sim g(\phi_{t-1})} \log p(y_{t-1}|x_{t-1}, \phi_{t-1}) \tag{16}$$

and employ the lower bound of $\log p(y_{t-1}|x_{t-1})$. Optimizing this lower bound generally produces satisfactory solutions, like in variational inference, VAE, and diffusion models.

**Rectification of** $1/\mathrm{E}(\phi_{t-1})$. By considering the diagonal Gaussian likelihood, the noise correlation between neighboring pixels in $y_0$ (and $y_t$) is ignored, which affects the estimation of the precision

**Algorithm 1** Difusion priors-based variational image denoising

---

**Input:** Pre-trained diffusion model, noisy observation $y_0$, hyperparameters $\alpha, \beta$, temperature $\gamma$
1: $x_T \sim \mathcal{N}(0, I), \mathrm{E}(\phi_T) = \vec{1}$
2: **for** $t = T, \cdots, 1$ **do**
3:     Compute $\mu_\theta(x_t, t)$ based on Eq. (4); Compute $y_{t-1}$ based on Eq. (7)
4:     Set $\hat{\mu}_{t-1}^{\mathrm{old}} = \vec{0}, \hat{\mu}_{t-1} = \mu_\theta(x_t, t)$
5:     **while** $\|\hat{\mu}_{t-1}^{\mathrm{old}} - \hat{\mu}_{t-1}\|_2^2 \geq 1e^{-6}$ **do**
6:         Update $g(x_{t-1}) = \mathcal{N}(\hat{\mu}_{t-1}, \hat{\sigma}_{t-1}^2)$ using Eq. (12)
7:         Update $g(\phi_{t-1}) = \prod_{i=1}^N \mathrm{Gamma}(\hat{\alpha}_{t-1}^i, \hat{\beta}_{t-1}^i)$ using Eq. (14)
8:     **end while**
9:     Solve optimal $x_{t-1}$ using Eq. (15) or Eq. (17)
10: **end for**
11: **return** $x_0$

---

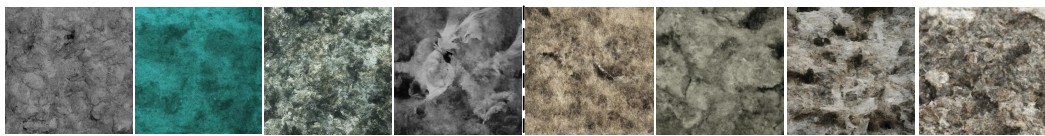

256×256 images sampled from 128×128 diffusion model ' 512×512 images sampled from 256×256 diffusion model

Figure 1: Unconditional HR images generation from LR diffusion models

posterior and noise variance $1/\mathrm{E}(\phi_{t-1})$. This is apparent as transformations in $f$ (e.g., demosaicking in the ISP pipeline) will cause the local correlation of noise variance (or precision) while elements in $\mathrm{E}(\phi_{t-1})$ are updated independently as shown in Eq. (14). Motivated by the analysis of spatial correlation of real-world noise in [23], we introduce a local 2D convolution with a normalized Gaussian kernel $G(l, s)$ ($l$ is kernel size and $s$ is the scale) to manually rectify $1/\mathrm{E}(\phi_{t-1})$. That is,

$$\overline{1/\mathrm{E}(\phi_{t-1})} = \mathrm{Conv}(1/\mathrm{E}(\phi_{t-1}), G(l, s)), \hat{\pi}_{t-1} = \frac{\sigma_t^2}{\sigma_t^2 + \overline{1/\mathrm{E}(\phi_{t-1})}} \tag{17}$$

By progressively calculating $x_{t-1}^*$ at each step $t$ based on Eqs. (15) and (17), the final denoised result $x_0$ can be obtained. The whole denoising algorithm is presented in Alg. 1.

### 3.4 Local Diffusion Priors

The common practice of sampling images from pre-trained diffusion models is to maintain the sampling resolution identical to the training resolution, which produces the best generation performance. In image restoration tasks, however, the resolution of the observed noisy image generally mismatches that of the pre-trained diffusion model. Existing approaches typically use patches [12] or resize operations [19] to address this issue, which is cumbersome and may affect local details.

We observe that when a diffusion model trained with the U-Net architecture on low-resolution (LR) images is employed to sample high-resolution (HR) images, it exhibits local properties. Fig. 1 shows sampled $512 \times 512$ and $256 \times 256$ images from pre-trained $256 \times 256$ [2] and $128 \times 128$ [3] diffusion models, respectively, and it is clear that the generated textures mainly focus on local areas. As HR images contain more redundant information, denoising HR images is simpler than denoising their LR counterparts under the same noise level [41]. When the LR diffusion model is used to generate HR samples, there is only a short time window for it to decide the structures of the sampled images [15], thus the generation tends to be local. Similar to traditional TV priors and Markov random fields that focus on designing local image statistics, we note that the local property of LR diffusion models is also effective for image denoising. This allows us to directly adopt the pre-trained LR diffusion prior to denoise HR noisy images without additional operations.

---

[2]https://openaipublic.blob.core.windows.net/diffusion/jul-2021/256x256_diffusion_uncond.pt
[3]https://openaipublic.blob.core.windows.net/diffusion/jul-2021/128x128_diffusion.pt

Table 1: Selection of $\beta$ and $s$ for different test datasets.

| Datasets | SIDD | FMDD | CC | PolyU | Datasets | SIDD | FMDD | CC | PolyU |
|---|---|---|---|---|---|---|---|---|---|
| $\beta$ value | 0.03 | 0.025 | 0.01 | 0.005 | $s$ value | 0.6 | 1.0 | 1.0 | 1.0 |

## 4 Experiments

### 4.1 Experimental settings

**Datasets**. We consider several real-world denoising datasets to evaluate our method, including SIDD [1], PolyU [47], CC [30], and FMDD [51]. SIDD validation, PolyU and CC datasets contain natural sRGB images from smartphones or commercial camera brands, where SIDD consists of 1280 patches with size $3 \times 256 \times 256$, PolyU and CC consist of 100 and 15 natural images with size $3 \times 512 \times 512$, respectively. FMDD contains 48 fluorescence microscopy images with size $512 \times 512$.

**Implementation Details**. We utilize the $256 \times 256$ unconditional diffusion model [2] provided by [9] as the diffusion prior throughout our main experiments, regardless of the resolution of input noisy images. The total diffusion steps are 1000 by default, i.e., $t \in [1, \cdots, 1000]$. We choose $\alpha = 1$ and Gaussian kernel size $l = 9$. The hyperparameters $\beta$ and $s$ for different datasets are summarized in Table 1. Different $\alpha/\beta$ represent the rough estimation of the prior precision for noises in different datasets, and Gaussian kernel scale $s$ controls the range of local spatial correlation. The temperature $\gamma$ is set to $1/5$ for all datasets and will be ablated in the sequel. For SIDD dataset, the sizes of $x_{0:T}$ are $3 \times 256 \times 256$. For the remaining datasets, they are $3 \times 512 \times 512$. The denoised results for FMDD are obtained by averaging the channel dimension of $x_0$ to get one-channel images. All experiments are conducted on Nvidia 2080Ti GPU.

**Compared methods.** We consider several representative single image-based unsupervised denoising methods, including DIP [42], Self2Self [36], PD-denoising [52], ZS-N2N [27], and ScoreDVI [7]. We also compare against AP-BSN [23], a self-supervised denoising method that can be trained on noisy images of test sets directly. In addition, several diffusion priors-based image restoration methods, including DDRM [19], GDP [12], and DR2 [45], are chosen to denoise real-world images. For these compared methods, we utilize their official source code and report the corresponding performance. Particularly, these diffusion-based methods employ the same diffusion prior [2] as ours. As DDRM can only handle $i.i.d.$ Gaussian noise and requires the noise std $\sigma_{\text{ddrm}}$, we hence set $\sigma_{\text{ddrm}} = \sqrt{\beta/\alpha}$, i.e., the prior noise std in our method. GDP introduced the hard data-fidelity term $\hat{s}\|\hat{x}_0 - y_0\|_2^2$, where $\hat{x}_0$ is the estimated denoised result at each reverse diffusion step, and $\hat{s}$ is the guidance scale. We tune different $\hat{s}$ for different datasets. DR2 first obtained the intermediate $x_{t-1}$ by adopting a low-pass filter $\Phi_D$ and setting $x_{t-1} = \Phi_D y_{t-1} + (1 - \Phi_D)x_{t-1}$ and then conducted inference from step $\tau + 0.25T$ to $\tau$. We set $D = 8$ (Downsampling factor) and $\tau = 100$ for DR2. We evaluate the quantitative denoising quality using PSNR and SSIM metrics.

### 4.2 Main Results

We present quantitative comparisons of different methods in Table 2 and visual comparisons in Figs. 2, 3 (and Figs. 5, 6, 7 in the Appendix A.4). Overall, our method achieves the best quantitative (*on average*) and qualitative performance across all compared methods.

First, while Self2Self excels on PolyU and CC datasets, it shows poor denoising capacity (both PSNR/SSIM values and visual results) on SIDD and FMDD that contain severe image noise, as shown in Fig. 5. PD-denoising is generally effective but often introduces small artifacts that hinder visual effects, as seen in Fig. 5. ZS-N2N struggles with real-world noisy images and typically leaves noticeable noise after denoising due to its reliance on the independent noise assumption. Regarding diffusion priors-based methods, GDP and GR2 perform poorly on real-world denoising, introducing artifacts (see Figs 5a and 7) and over-smoothing (see Fig. 3), possibly because they adopt hard data-consistency methods, which significantly deviate from the true likelihood function. As DDRM assumes Gaussian white noise, the denoised images often retain some noise, which cannot be entirely removed, as shown in Figs 3 and 5b. ScoreDVI is the most competitive method against ours, but it sometimes blurs images and loses local textures, as indicated by Fig. 2. Unlike these methods, our approach effectively removes severe noise while preserving image details and textures.

Table 2: Quantitative comparisons (PSNR(dB)/SSIM) of different methods on diverse real-world image datasets. The best and second-best PSNR/SSIM results are marked in **bold** and underlined.

| Methods | SIDD Validation [1] | FMDD [51] | PolyU [47] | CC [30] | Average |
|---------|---------------------|-----------|------------|---------|---------|
| DIP [42] | 32.11/0.740 | 32.90/0.854 | 37.17/0.912 | 35.61/0.912 | 34.45/0.855 |
| Self2Self [36] | 29.46/0.595 | 30.76/0.695 | 38.33/0.962 | 37.45/0.948 | 34.00/0.800 |
| PD-denoising [52] | 33.97/0.820 | 33.01/0.856 | 37.04/0.940 | 35.85/0.923 | 34.97/0.885 |
| ZS-N2N [27] | 25.58/0.433 | 31.61/0.767 | 36.05/0.916 | 33.58/0.854 | 31.71/0.743 |
| ScoreDVI [7] | 34.75/0.856 | 33.10/**0.865** | 37.77/0.959 | 37.09/0.945 | 35.68/0.906 |
| GDP [12] | 27.65/0.615 | 27.68/0.698 | 32.30/0.905 | 31.45/0.916 | 29.77/0.784 |
| DR2 [45] | 32.02/0.728 | 30.52/0.813 | 34.37/0.925 | 32.30/0.876 | 32.30/0.836 |
| DDRM [19] | 33.14/0.796 | 32.54/0.837 | 33.14/0.767 | 36.04/0.923 | 33.72/0.831 |
| Ours | 34.76/**0.887** | **33.14**/0.860 | **38.71/0.970** | **38.01/0.959** | **36.16/0.919** |
| APBSN [23] | **36.80**/0.874 | 31.99/0.836 | 37.03/0.951 | 34.88/0.925 | 35.18/0.897 |

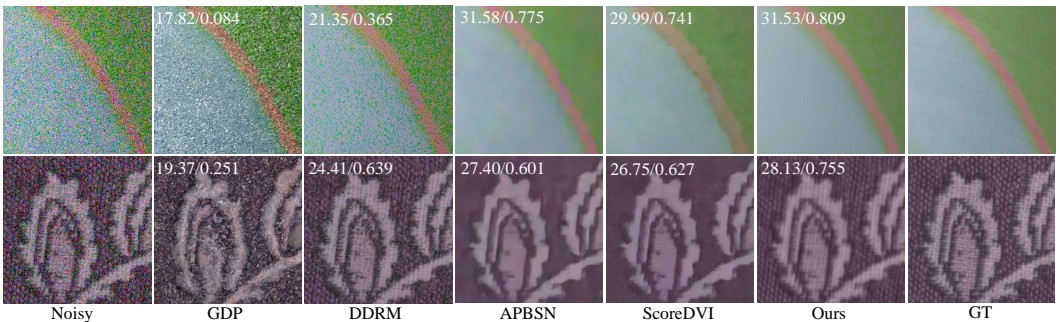

Figure 2: Visual comparison of different denoising methods in SIDD validation dataset.

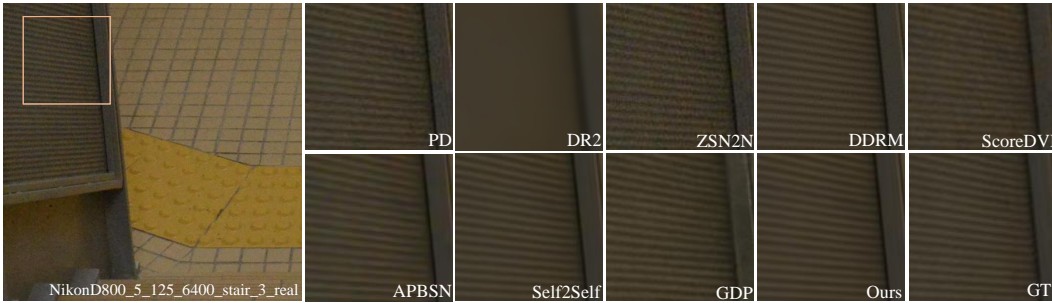

Figure 3: Visual comparison of different denoising methods in PolyU. PSNR/SSIM values: PD (36.77/0.916), APBSN (37.59/0.944), DR2 (34.53/0.864), Self2Self (39.44/0.961), ZSN2N (35.12/0.879), GDP (33.43/0.888), DDRM (33.61/0.773), ScoreDVI (37.76/0.939), Ours (39.01/0.965)

Although APBSN achieves the best PSNR values in the SIDD dataset, it frequently introduces noticeable color artifacts (see Figs. 2, 3) and oversmooths images (see Fig. 7). In addition, when applied to FMDD, PolyU, and CC datasets that contain fewer noisy images, its denoising performance significantly degrades and underperforms our method. This highlights the advantage of our data-efficient approach.

### 4.3 Ablation and Analyses

**Adaptive likelihood estimation (ALE).** We analyze $1/\mathrm{E}(\phi_0) = \hat{\beta}_0/\hat{\alpha}_0$, the estimated noise variance, and present quantitative and qualitative results in Fig. 4. Fig. 4a demonstrates that $\hat{\beta}_0/\hat{\alpha}_0$ effectively reflects the noise variance of $y_0$. That is, $\hat{\beta}_0/\hat{\alpha}_0$ exhibits larger values in noisier areas of $y_0$ and smaller values (black) in less noisy areas. Fig. 4b implies that the average of $\hat{\beta}_0/\hat{\alpha}_0$ are inversely correlated with PSNR values of denoised images, which is reasonable since noisier images are more challenging to denoise and hence have lower PSNR. Conversely, the prior noise variance $\beta/\alpha = 3e^{-3}$,

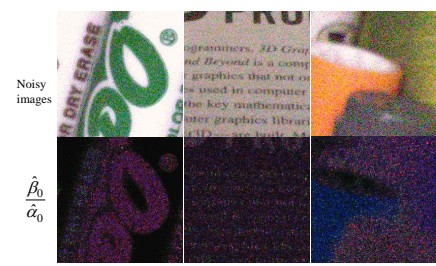 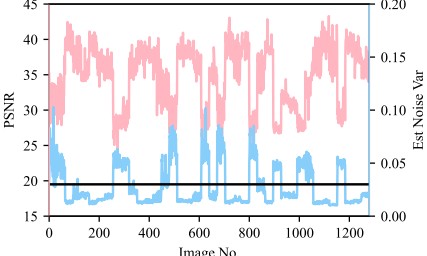

| (a) Visual results of $\hat{\beta}_0/\hat{\alpha}_0$ | (b) PSNR vs average $\hat{\beta}_0/\hat{\alpha}_0$. |

Figure 4: The estimated noise variance $1/\mathrm{E}(\phi_0) = \hat{\beta}_0/\hat{\alpha}_0$ on SIDD dataset

Table 3: Ablation on adaptive likelihood estimation and local Gaussian convolution

| with ALE | with Gaussian Conv | SIDD | FMDD | PolyU | CC |
|---|---|---|---|---|---|
| ✗ | ✗ | 32.12/0.741 | 27.07/0.530 | 35.40/0.895 | 33.10/0.830 |
| ✓ | ✗ | 34.63/0.870 | 33.11/**0.865** | 38.70/0.969 | 37.82/0.956 |
| ✓ | ✓ | **34.76/0.887** | **33.14**/0.860 | **38.71/0.970** | **38.01/0.959** |

indicated by the black plot in Fig. 4b are constant for all noisy images. These analyses suggest that the ALE captures the signal-dependent features of real-world noise effectively.

In addition, we skip the variational inference process and directly use the prior precision $p(\phi_{t-1})$ to derive $x_{t-1}^*$, resulting in $x_{t-1}^* = \pi_{t-1}y_{t-1} + (1 - \pi_{t-1})\mu_\theta(x_t, t), \pi_{t-1} = \sigma_t^2\alpha/(\sigma_t^2\alpha + \beta)$. The corresponding denoising result (i.e., without ALE) is reported in the second row of Table 3 and is largely behind the denoising result of using ALE, further verifying the effectiveness of ALE.

**Rectification in Eq.** (17). By introducing the local Gaussian convolution operation, we explicitly refine the estimated noise variance $1/\mathrm{E}(\phi_{t-1})$. As shown in the fourth row of Table 3, using $\overline{1/\mathrm{E}(\phi_{t-1})}$ consistently improves the quantitative performance.

Table 4: Ablation of temperature $\gamma$ on CC dataset

| $\gamma$ value | 1 | 1/2 | 1/4 |
|---|---|---|---|
| PSNR/SSIM | 37.74/0.955 | 37.73/0.955 | 37.90/0.957 |
| $\gamma$ value | 1/5 | 1/10 | 1/20 |
| PSNR/SSIM | **38.01/0.959** | 37.80/0.953 | 35.55/0.913 |

Table 5: Ablation of $\beta$ and $s$ on CC dataset

| $\beta$ value | 5e-3 | 1e-2 | 1.5e-2 |
|---|---|---|---|
| PSNR/SSIM | 38.03/0.957 | 38.01/0.959 | 37.47/0.953 |
| $s$ value | 0.8 | 1.0 | 1.2 |
| PSNR/SSIM | 37.876/0.957 | 38.01/0.959 | 38.10/0.959 |

Table 6: Denoising performance of using diffusion priors pre-trained with other image resolutions

| Res.: Train $\rightarrow$ Test | SIDD | Res.: Train $\rightarrow$ Test | CC | PolyU | FMDD |
|---|---|---|---|---|---|
| $128 \rightarrow 256$ | **34.80**/0.836 | $256 \rightarrow 512$ | **38.01/0.959** | **38.71/0.970** | **33.14/0.860** |
| $256 \rightarrow 256$ | 34.76/**0.887** | $512 \rightarrow 512$ | 37.01/0.950 | 38.33/0.966 | 33.02/0.859 |

**Temperature** $\gamma$. When $\gamma \leq 1$, the effect of diffusion priors $p(x_{t-1}|x_t)$ is reduced within the variational inference during the reverse diffusion process. As shown in Table 4, decreasing $\gamma$ gradually improves the quantitative denoising performance, peaking at $\gamma = 1/5$. Further reducing $\gamma$ degrades PSNR/SSIM as insufficient diffusion priors are involved in variational Bayes.

$\beta$ **in prior precision and kernel scale** $s$. Regarding $\beta$, it roughly represents the noise level of noisy image $y$ (given $\alpha = 1$) and we choose $\beta$ according to the empirical variance of the textureless area of $y$ for one test set. kernel scale $s$ is set by considering the spatial correlation of noise present in real-world images. We ablate these two parameters in Table 5, which indicates that they are relatively robust to moderate changes.

**Local diffusion priors**. We consider diffusion models pre-trained with other image resolutions as diffusion priors, including the $128 \times 128$ version and $512 \times 512$ version [4]. The corresponding denoising performance is reported in Table 6, and implementation details are provided in the Appendix A.3.1. Regarding SIDD, we observe that matching the resolution of diffusion priors and test images (both $256 \times 256$) achieves the best performance. Such a result is reversed for the remaining datasets,

---

[4]https://openaipublic.blob.core.windows.net/diffusion/jul-2021/512x512_diffusion.pt

Table 7: The quantitative performance on removing other non-Gaussian synthetic noises

| CBSD68 [28] | Poisson ($\lambda = 30$) | Bernoulli ($p = 0.2$) | KodaK [13] | Poisson ($\lambda = 30$) | Bernoulli ($p = 0.2$) |
|---|---|---|---|---|---|
| ZS-N2N | 27.55/0.781 | 20.20/**0.828** | ZS-N2N | 28.09/0.750 | 19.98/**0.820** |
| Ours | **29.24/0.833** | **26.11**/0.784 | Ours | **30.56/0.839** | **27.17**/0.799 |

where the $256 \times 256$ diffusion prior for HR images is more effective than its $512 \times 512$ counterpart. This is likely because training HR diffusion models (e.g., $\geq 512$) is more challenging, resulting in inferior generation performance (e.g., FID) compared to LR diffusion models [9]. Consequently, the local prior inherent in medium-resolution diffusion models is superior for restoring HR images.

### 4.4 Application to other non-Gaussian noises

Although our method is designed to handle real-world noise, it can also address other non-Gaussian noises, including Poisson noise and multiplicative Bernoulli noise. As these synthetic noises are spatially independent, we do not utilize Eq. (17) in our method. We report the denoising performance in Table 7, with ZS-N2N [16] selected for comparison. Experimental details and visual results are given in the Appendix A.3.2 and A.5. Our method achieves better quantitative metrics than ZS-N2N on Poisson denoising and also preserves more local details and textures (see Fig. 8). While ZS-N2N shows better SSIM than ours on Bernoulli denoising, it causes intensity shifts (see Fig. 9) and thus has poorer PSNR.

### 4.5 Application to image demosaicing

In addition to denoising, our method is readily available for image restoration with pixel-wise degradation, e.g., image demosaicing. To adapt our method to this task, we define the forward process $y_0 = M \odot x_0$, where $M$ is the degradation operator, and denotes element-wise multiplication. For demosaicing, $M$ is the binary mask with 0 values indicating missing pixels of $y_0$. We can incorporate $M$ into $p(y_{t-1}|x_{t-1}, \phi_{t-1})$ in Eq. (15), which results in $\hat{\pi}_{t-1} = \frac{M\sigma_t^2}{M\sigma_t^2 + 1/\mathrm{E}(\phi_{t-1})}$. In Table 8, we compare our method against DDRM on image demosaicing (CFA pattern: RGGB), and our method shows better results.

Table 8: Results of image demosaicing

| Dataset | Set14 | CBSD68 |
|---|---|---|
| DDRM | 24.68/0.714 | 24.52/0.705 |
| Ours | **26.02/0.756** | **25.43/0.732** |

Table 9: Results of different sampling steps

| Step | 1000 | 500 | 250 |
|---|---|---|---|
| SIDD Val | **34.76/0.887** | 33.54/0.838 | 23.89/0.825 |
| CC | **38.01/0.959** | 37.18/0.947 | 22.72/0.716 |

**Limitation**. Our method builds on the DDPM sampling with a total of 1000 diffusion steps. Denoising a single noisy image with a resolution of $256 \times 256$ on an Nvidia 2080Ti GPU takes approximately 230 seconds, which is inefficient. In comparison, ZS-N2N takes about 16 seconds, despite its inferior denoising performance. Naively reducing diffusion steps in our method leads to apparent performance decreases, as indicated in Table 9. Our next move is to incorporate advanced accelerated sampling strategies into our method to reduce inference time while maintaining performance.

## 5 Conclusion

In this paper, we built upon diffusion priors and variational Bayes and proposed adaptive likelihood estimation and MAP inference during the reverse diffusion process, to handle real-world image noise that is structured and signal-dependent. The employed $i.ni.d.$ likelihood function, combined with the precision prior and variational Bayes, allowed for the dynamical update of $i.ni.d.$ noise precision posterior in each step of the generation process. This strategy adaptively refined the likelihood function and enabled the better MAP inference. Our method achieved excellent denoising performance on diverse real-world image denoising datasets and was also effective for removing other non-Gaussian synthetic noises.

## Acknowledgment

This work was supported in part by the National Natural Science Foundation of China (NNSFC), under Grant Nos. 61672253, 62071197, and 62471192.

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

# A Appendix

## A.1 Derivation of Eq. (6)

As analyzed in Section 3.2, in principle we can model the likelihood function of real-world noisy images as $p(y_0|x_0) = \mathcal{N}(x_0, \Sigma(x_0))$, where $\Sigma$ is a non-diagonal covariance matrix and its variance is related to its mean $x_0$ (or signal). In order to incorporate $y_0$ into the inverse diffusion process and shorten its gap to $x_{t-1}$, we can construct $y_{t-1}$ based Eq. (2) to obtain $y_{t-1} = \sqrt{\bar{\alpha}_{t-1}}y_0 + \sqrt{1 - \bar{\alpha}_{t-1}}\epsilon_2$. For $y_0$, it can be sampled from the multi-variate Gaussian $p(y_0|x_0) = \mathcal{N}(x_0, \Sigma(x_0))$, i.e., $y_0 = x_0 + A\epsilon$, where $AA^T = \Sigma(x_0)$, and $A$ is obtained by Cholesky decomposition. Finally, we obtain

$$
\begin{aligned}
y_{t-1} &= \sqrt{\bar{\alpha}_{t-1}}y_0 + \sqrt{1 - \bar{\alpha}_{t-1}}\epsilon = \sqrt{\bar{\alpha}_{t-1}}(x_0 + A\epsilon_2) + \sqrt{1 - \bar{\alpha}_{t-1}}\epsilon \\
&= \sqrt{\bar{\alpha}_{t-1}}x_0 + \sqrt{1 - \bar{\alpha}_{t-1}}\epsilon + \sqrt{\bar{\alpha}_{t-1}}A\epsilon_2 = x_{t-1} + \sqrt{\bar{\alpha}_{t-1}}A\epsilon_2
\end{aligned}
\tag{18}
$$

## A.2 Derivations of Eqs. (11) and (13)

Regarding Eq. (10), we have

$$
\begin{aligned}
&E_{\phi_{t-1}} \log p(y_{t-1}|x_{t-1}, \phi_{t-1})^{\frac{1}{\gamma}} p(x_{t-1}|x_t)p(\phi_{t-1}) \\
&= E_{\phi_{t-1}} \sum_{i=1}^{N} \left( -\frac{(y_{t-1}^i - x_{t-1}^i)^2}{2\gamma}\phi_{t-1}^i - \frac{(x_{t-1}^i - \mu_t^i)^2}{2\sigma_t^2} \right) + \text{const} \\
&= \sum_{i=1}^{N} \left( -\frac{(y_{t-1}^i - x_{t-1}^i)^2}{2\gamma}E(\phi_{t-1}^i) - \frac{(x_{t-1}^i - \mu_t^i)^2}{2\sigma_t^2} \right) + \text{const} \\
&= \sum_{i=1}^{N} -\frac{(y_{t-1}^i - x_{t-1}^i)^2 E(\phi_{t-1}^i)\sigma_t^2 + (x_{t-1}^i - \mu_t^i)^2\gamma}{2\gamma\sigma_t^2} + \text{const} \\
&= \sum_{i=1}^{N} -\frac{(E(\phi_{t-1}^i)\sigma_t^2 + \gamma)(x_{t-1}^i)^2 - 2(E(\phi_{t-1}^i)\sigma_t^2 y_{t-1}^i + \mu_t^i\gamma)x_{t-1}^i}{2\gamma\sigma_t^2} + \text{const}
\end{aligned}
\tag{19}
$$

where $\mu_t = \mu_\theta(x_t, t)$. We observe that Eq. (19) has the summation and quadratic form of $x_{t-1}^i$, and hence $g(x_{t-1})$ is identified as a diagonal Gaussian distribution. By completing the square, we can obtain

$$
g(x_{t-1}) = \mathcal{N}\left( \frac{\sigma_t^2 E(\phi_{t-1}) \odot y_{t-1} + \mu_t\gamma}{E(\phi_{t-1})\sigma_t^2 + \gamma}, \text{diag}\left( \frac{\gamma\sigma_t^2}{E(\phi_{t-1})\sigma_t^2 + \gamma} \right) \right)
\tag{20}
$$

Similarly, given $g(x_{t-1})$, the optimal $g^*(\phi_{t-1})$ is provided by

$$
\log g^*(\phi_{t-1}) = E_{x_{t-1}} \log p(y_{t-1}|x_{t-1}, \phi_{t-1})^{\frac{1}{\gamma}} p(x_{t-1}|x_t)p(\phi_{t-1})
\tag{21}
$$

which corresponds to

$$
\begin{aligned}
&E_{x_{t-1}} \log p(y_{t-1}|x_{t-1}, \phi_{t-1})^{\frac{1}{\gamma}} p(x_{t-1}|x_t)p(\phi_{t-1}) \\
&= E_{x_{t-1}} \sum_{i=1}^{N} \left( \frac{1}{2\gamma}\log\phi_{t-1}^i - \frac{(y_{t-1}^i - x_{t-1}^i)^2}{2\gamma}\phi_{t-1}^i + (\alpha_{t-1} - 1)\log\phi_{t-1}^i - \beta_{t-1}\phi_{t-1}^i \right) + \text{const} \\
&= \sum_{i=1}^{N} \left( \frac{1}{2\gamma}\log\phi_{t-1}^i - \frac{(y_{t-1}^i - E(x_{t-1}^i))^2 + \sigma_t^2}{2\gamma}\phi_{t-1}^i + (\alpha_{t-1} - 1)\log\phi_{t-1}^i - \beta_{t-1}\phi_{t-1}^i \right) + \text{const}
\end{aligned}
\tag{22}
$$

From Eq. (22), we identify each $g(\phi_{t-1}^i)$ is a independent gamma distribution and hence $g(\phi_{t-1})$ is

$$
g^*(\phi_{t-1}) = \prod_{i=1}^{N} \text{Gamma}\left( \phi_{t-1}^i; \alpha_{t-1} + \frac{1}{2\gamma}, \beta_{t-1} + \frac{(y_{t-1}^i - \hat{\mu}_{t-1}^i)^2 + (\hat{\sigma}_{t-1}^2)^i}{2\gamma} \right)
\tag{23}
$$

## A.3 Additional implementation details

### A.3.1 Applying class-conditional diffusion models as diffusion priors

We note that [9] only provides the $256 \times 256$ unconditional diffusion model $\epsilon_\theta(x_t, t)$ trained on ImageNet, and the remaining diffusion models are class-conditional, i.e., $\epsilon_\theta(x_t, c, t)$ with the class label $c$. There are two ways to utilize these class-conditional diffusion priors for image denoising. Regarding each noisy image, we first sample a class label $c$ from $randint(0, 1000)$ and input it to the $\epsilon_\theta(x_t, c, t)$ combined with $x_t$ and $t$. One way is then ignoring the guidance from the pre-trained classifier during the generation process and directly computing $\mu_\theta(x_t, t)$ based on $\epsilon_\theta(x_t, c, t)$ and Eq. (4). The other way is further updating $\mu_\theta(x_t, t)$ to $\mu_\theta(x_t, t) + g_s \sigma_t^2 \nabla_{x_t} \log p_\psi(c|x_t)$, where $p_\psi(c|x_t)$ is the classifier and $g_s$ is the guidance scale. Basically, we found these two ways resulted in similar denoising performance, and Table 6 used the first way.

### A.3.2 Experiments on denoising non-Gaussian synthetic noises

**Synthesis of noisy images**. Regarding Bernoulli noise, we obtain the noisy image by $y_0 = x_0 \odot M$, $M = \text{torch.bernoulli}(\text{torch.ones\_like}(x_0) * p)$, $p = 0.2$; Regarding Poisson noise, we obtain the noisy image by $y_0 = \text{torch.poisson}(\lambda * x_0)/\lambda$, $\lambda = 30$.

**Hyperparameters**. For our method, we set $\beta = 1e^{-2}$ and $\beta = 8e^{-3}$ for Bernoulli noise and Poisson noise removal, respectively. The remaining hyperparameters are identical to those of our main experiments.

### A.4 Visual comparisons of denoising results on real-world datasets

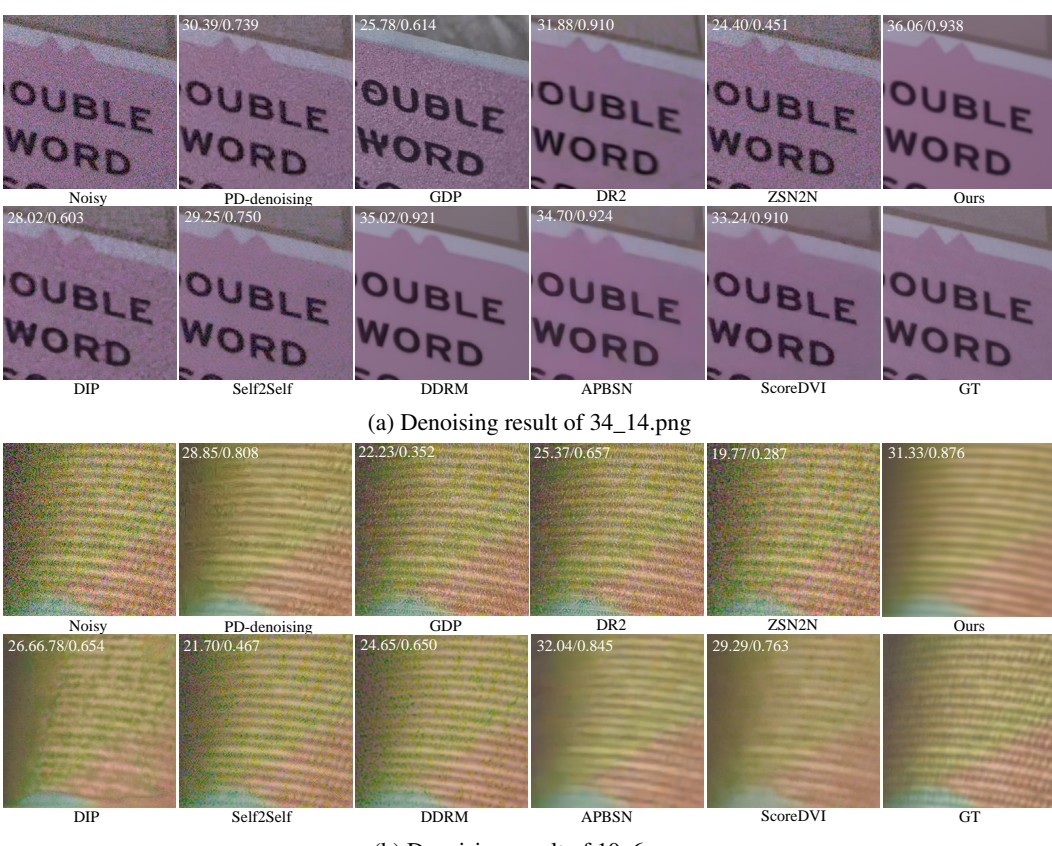

(a) Denoising result of 34_14.png

(b) Denoising result of 10_6.png

Figure 5: Visual comparison of different denoising methods in SIDD validation dataset.

### A.5 Visual comparisons of denoising results on synthetic noises

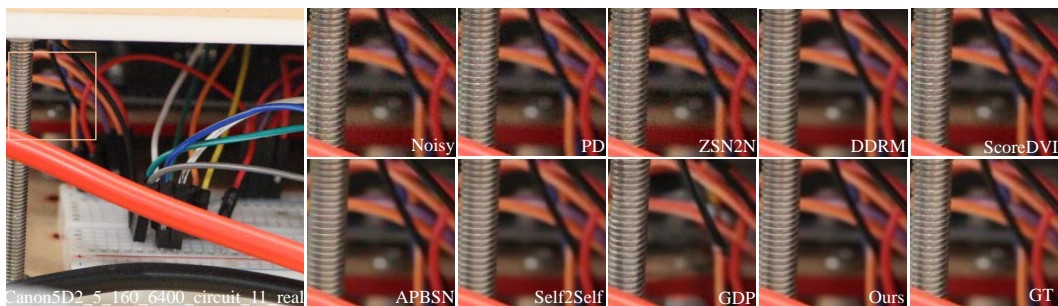

Figure 6: Visual comparison of different denoising methods in PolyU.

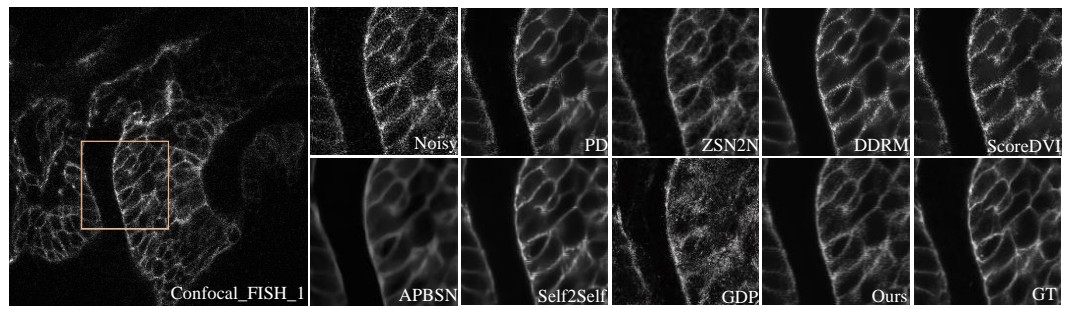

Figure 7: Visual comparison of different denoising methods in FMDD.

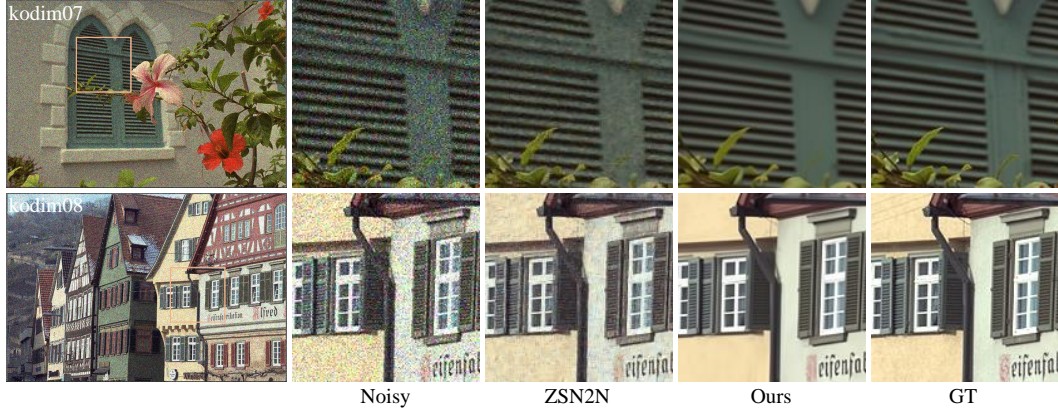

Figure 8: Visual comparison of denoising results on Poisson denoising ($\lambda = 30$)

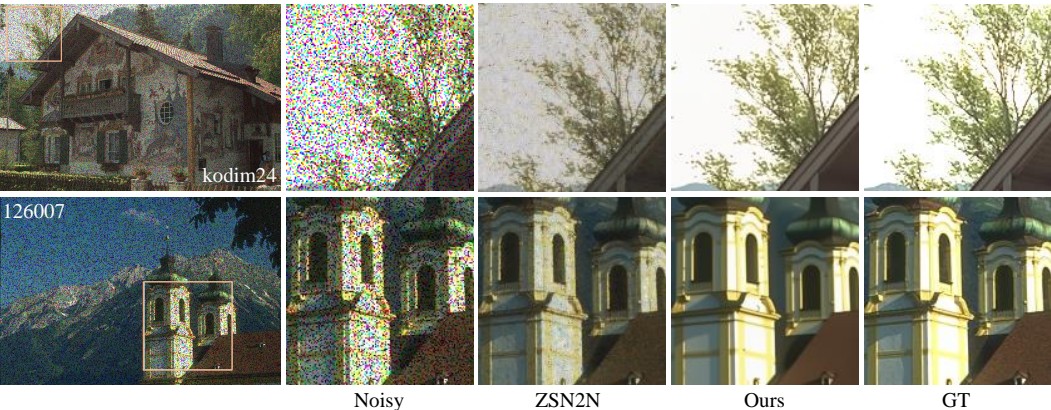

Figure 9: Visual comparison of denoising results on Bernoulli denoising ($p = 0.2$)

