# OpenReview forum: "Diffusion Priors for Variational Likelihood Estimation and Image Denoising"
_NeurIPS.cc/2024/Conference — NeurIPS 2024 spotlight_

### Official Review · Reviewer_Wzuv · 2024-06-17

**Soundness:** 3
**Presentation:** 3
**Contribution:** 3
**Rating:** 6
**Confidence:** 4

**Summary:**

This paper proposed a diffusion-based image denoising method. The method leverages the MAP framework with proposed adaptive likelihood estimation method and a pre-trained diffusion prior. Experiments on four real-world datasets validate the advantages of the proposed method.

**Strengths:**

1. The paper is well-written and easy to understand.
2. The integration of a reverse diffusion process with adaptive likelihood estimation is novel.
3. The denoising performance of the proposed method outperforms that of other single-image denoising methods. The experimental results are substantial.

**Weaknesses:**

1. The title of the paper does not accurately reflect its contributions. The use of diffusion priors to address inverse problems is widely studied, as the author has noted, and this paper builds upon the framework presented in [1]. The significant contribution of this paper appears to be the proposed alternative updating scheme for estimating the likelihood term. However, the title is too generic and does not capture this contribution.
2. The denoising computational cost is high, and the running time for the reverse diffusion process is lengthy.
3. In Algorithm 1: Difusion -> Diffusion.

**Questions:**

The proposed algorithm involves numerous hyperparameters. How are these selected, and is the algorithm robust to variations in hyperparameter settings?

**Limitations:**

The authors have acknowledged the limitations of their work.

---

> ### Author Rebuttal · Authors · 2024-08-04
>
> Thanks for your recognition of our work.
>
> Q1: The title of the paper does not accurately reflect its contributions. The use of diffusion priors to address inverse problems is widely studied, as the author has noted, and this paper builds upon the framework presented in [1]. The significant contribution of this paper appears to be the proposed alternative updating scheme for estimating the likelihood term. However, the title is too generic and does not capture this contribution.
>
> Reply: Thanks for your suggestion, and we would like to modify our title to the following one in the revised version if allowed:
> "Diffusion priors for variational likelihood estimation and image denoising".
>
> Q2: The denoising computational cost is high, and the running time for the reverse diffusion process is lengthy.
>
> Reply: As pointed out in the limitation of the main paper, our method relies on DDPM and large sampling steps for good denoising performance. We tried several other sampling steps and report the performance in the following Table:
>
> |Steps     | $T=1000$ |  $T=500$  | $T=250$  |
> | ----------- | ----------- | ----------- | ----------- |
> | SIDD Val      | 34.76/0.887  | 33.54/0.838  | 23.89/0.825   |
> | CC      | 38.01/0.959     | 37.18/0.947   | 22.72/0.716  |
>
> It can be observed that $T=500$ degrades the performance, which is acceptable though. In the case of $T=250$, the denoising completely fails. As a result, speeding up the inference is our key direction in the future.
>
> Q3: In Algorithm 1: Difusion -> Diffusion
>
> Reply: Thanks for your careful review, and we will correct it in the revised version.
>
> Q4: The proposed algorithm involves numerous hyperparameters. How are these selected, and is the algorithm robust to variations in hyperparameter settings?
>
> Reply: Our method involves three hyperparameters: $\beta$ in the noise prior precision, temperature $\gamma$ and kernel scale $s$. Regarding $\beta$ value, it roughly represents the noise level of noisy image $y$ (given $\alpha=1$) and we can choose $\beta$ according to the empirical variance of the textureless area of $y$ for one testset. kernel scale $s$ is set by considering the spatial correlation of noise present in real-world images. Temperature $\gamma$ is an empirical parameter that scales the likelihood function in Bayes' theorem and is generally set to values smaller than 1. We ablate these three parameters in the following tables regarding the CC dataset, which indicate that they are relatively robust to moderate changes.
>
> | $\beta$($\gamma=5,s=1$) | 5e-3 | 1e-2 | 1.5e-2 |
> | :--- |  :---: | :---: | :---:  |
> | PSNR/SSIM |38.03/0.957| 38.01/0.959 | 37.47/0.953|
>
> | $\gamma$($\beta=0.01,s=1$) | 1/4 | 1/5 | 1/10 |
> | :--- | :---: | :---: | :---: |
> | PSNR/SSIM| 37.90/0.957| 38.01/0.959| 37.80/0.953|
>
> | $s$($\gamma=5,\beta=0.01$)| 0.8 | 1.0 | 1.2 |
> | :--- | :---: | :---: | :---: |
> | PSNR/SSIM| 37.876/0.957| 38.01/0.959|38.10/0.959|

---

> > ### Comment · Reviewer_Wzuv · 2024-08-12
> > **Thanks for the response.**
> >
> > Thank you for your response. Regarding using fewer sampling steps, are you referring to the DDIM sampler? Additionally, utilizing advanced diffusion samplers, such as DPM-Solver, may help accelerate the sampling speed. I have no further concerns and will maintain my score.

---

> > > ### Author Response · Authors · 2024-08-12
> > >
> > > Thanks for your reply and suggestion. Regarding using fewer sampling steps in the response to Q2, we used the DDPM sampler with fewer steps (e.g., $T=500$ means taking one sample per two steps compared with original $T=1000$). As you suggested, we will try to incorporate DDIM or DPM-Solver into our method to improve the sampling efficiency while maintaining the denoising performance.

---

### Official Review · Reviewer_sawX · 2024-07-09

**Soundness:** 3
**Presentation:** 3
**Contribution:** 3
**Rating:** 6
**Confidence:** 4

**Summary:**

The authors propose a way to use diffusion priors for real-world image denoising where the noise statistics are complex and signal-dependent. They use variational inference to estimate the joint posterior of the noise precision and image throughout diffusion time. The result is a MAP estimate of the denoised image that takes into account signal dependence and spatial correlation of the observed noise. The authors also propose to use a low-resolution diffusion model to denoise high-resolution images. In experiments, the authors compare to single image-based denoising methods, a self-supervised denoising method, and diffusion-based methods. They show improved quantitative and qualitative performance against the baselines.

**Strengths:**

* The proposed method is sound and addresses a difficult problem.
* Plenty of baselines are provided, and experiments show convincing quantitative and qualitative performance. Plus a good number of ablation studies is provided.
* The proposed method works for other non-Gaussian synthetic noise as well as real-world camera noise.

**Weaknesses:**

* The proposed method only provides MAP estimates (not posterior samples).
* Nit: Eq. 6 should be explained more clearly in the text. For example, A should be explicitly defined. And why is the same $\epsilon$ used twice? Shouldn’t it be $y_{t-1}=\sqrt{\bar{\alpha}_{t-1}}(x_0+A\epsilon_1)+\sqrt{1-\bar{\alpha}_{t-1}}\epsilon_2$ for $\epsilon_1,\epsilon_2$ independently drawn (since the measured noise is different from the synthetic noise added in the diffusion process)?

**Questions:**

* How difficult would it be to adapt this method to provide posterior samples?

**Limitations:**

I appreciate the authors providing a limitations section. There are probably more limitations of the method that could be discussed, such as not addressing posterior sampling.

---

> ### Author Rebuttal · Authors · 2024-08-04
>
> Thanks for your recognition of our work. As some questions overlap with the weaknesses, we will integrate and answer them together.
>
> Q1: The proposed method only provides MAP estimates (not posterior samples). How difficult would it be to adapt this method to provide posterior samples?
>
> Reply: Employing MAP inference is more straightforward for our method and also reduces the sampling randomness. In order to obtain posterior samples, we have to sample $x_{t-1}$ from $p(x_{t-1}|y_{t-1},x_t)$ rather than derive its MAP solution as done in Eq. (15). For our method, it is not very difficult to achieve that. First, we can approximate $p(y_{t-1}|x_{t-1})$ by  Monte-Carlo integration, resulting in
> $$p(y_{t-1}|x_{t-1})= E_{g(\phi_{t-1})}p(y_{t-1}|x_{t-1}, \phi_{t-1}) \approx \frac{1}{M}\sum_{s=1}^Mp(y_{t-1}|x_{t-1}, \phi^s_{t-1}) \approx \mathcal{N}(y_{t-1};x_{t-1}, \frac{1}{M}\sum_{s=1}^M (\phi^s_{t-1})^{-1})$$
> where we utilize uni-Gaussian to approximate the mixture Gaussian regarding the last term; $\phi^s_{t-1} \sim g(\phi_{t-1})$; $M$ is the number of Mente-Carlo samples and should be as large as possible.
>
> Then, as both $p(y_{t-1}|x_{t-1})$ and $p(x_{t-1}|x_t)$ follow Gaussian distributions, $p(x_{t-1}|y_{t-1},x_t)$ is also the Gaussian distribution, from which we can sample $x_{t-1}$  and the final $x_0$.
>
> We provide some posterior samples of the denoising result in Figure 3 of the 6958_rebuttal.pdf for your reference. We would like to discuss this posterior sampling variant in the revised paper.
>
> Q2: Nit: Eq. 6 should be explained more clearly in the text. For example, A should be explicitly defined. And why is the same ϵ used twice? Shouldn’t it be $y_ {t-1} =\sqrt{\bar{\alpha}_ {t-1}}(x_0+A\epsilon_1)+\sqrt{1-\bar{\alpha}_ {t-1}}\epsilon_2$ for $\epsilon_1$,$\epsilon_2$ independently drawn (since the measured noise is different from the synthetic noise added in the diffusion process)?
>
> Reply: First, we are grateful for pointing out the mistake in Eq. (6), and it should be
>
> $$y_ {t-1}=\sqrt{\bar{\alpha}_ {t-1}}(x_0+A\epsilon_1)+\sqrt{1-\bar{\alpha}_ {t-1}}\epsilon_2=x_ {t-1}+\sqrt{\bar{\alpha}_ {t-1}}A\epsilon_1, \epsilon_1 \sim \mathcal{N}(0, I), \epsilon_2 \sim \mathcal{N}(0, I)$$
>
> In the following, we will give a detailed derivation and explanation of this equation. As analyzed in Section 3.2, in principle we can model the likelihood function of real-world noisy images as $p(y_0|x_0)=\mathcal{N}(x_0, \Sigma(x_0))$, where $\Sigma$ is a non-diagonal covariance matrix and its variance is related to its mean $x_0$ (or signal). In order to incorporate $y_0$ into the inverse diffusion process and shorten its gap to $x_{t-1}$, we can construct $y_{t-1}$ based Eq. (2) to obtain $y_ {t-1} = \sqrt{\bar{\alpha}_ {t-1}}y_0 + \sqrt{1-\bar{\alpha}_ {t-1}}\epsilon_2$. For $y_0$, it can be sampled from the multi-variate Gaussian $p(y_0|x_0)=\mathcal{N}(x_0, \Sigma(x_0))$, i.e., $y_0=x_0+A\epsilon_1$, where $AA^T=\Sigma(x_0)$, and $A$ is obtained by Cholesky decomposition. Finally, we obtain
> $$y_ {t-1} = \sqrt{\bar{\alpha}_ {t-1}}y_0 + \sqrt{1-\bar{\alpha}_ {t-1}}\epsilon_2 = \sqrt{\bar{\alpha}_ {t-1}} (x_0+A\epsilon_1) + \sqrt{1-\bar{\alpha}_ {t-1}}\epsilon_2 = \sqrt{\bar{\alpha}_ {t-1}} x_0 + \sqrt{1-\bar{\alpha}_ {t-1}}\epsilon_2 + \sqrt{\bar{\alpha}_ {t-1}} A\epsilon_1 = x_ {t-1} + \sqrt{\bar{\alpha}_ {t-1}} A\epsilon_1 $$
>
> We will add the above explanation of Eq. (6) in the revised paper.

---

> > ### Comment · Reviewer_sawX · 2024-08-13
> >
> > Thank you to the authors for their response. It would be good to add the explanation of Eq. 6 to the revised paper, but I would prefer not to add the posterior sampling variant. I would want to see further explanation/experiments to verify the proposed approximation of the posterior. I appreciate the authors' efforts at extending their method to posterior sampling nevertheless. I have no further questions/concerns and maintain my rating.

---

### Official Review · Reviewer_tZQC · 2024-07-17

**Soundness:** 2
**Presentation:** 2
**Contribution:** 2
**Rating:** 5
**Confidence:** 3

**Summary:**

This work considers the problem of using adapt diffusion models for solving real-world image denoising problem, that is, the noise is not assumed to be i.i.d Gaussian. The authors statistically model the real-world noise as independent, non-identically distribution noise, and then incorporate the adaptive MAP inference into the reverse process of the diffusion model.

**Strengths:**

N/A

**Weaknesses:**

1. The paper does not well motivate their choice of i.ni.d noise model. It is not clear why such an i.ni.d noise model can properly characterize the statistics of the real-world noise considered in the paper.
2. The paper is not well-written, lacking proper motivation and explanation of the design of the proposed method.
3. The approximation adopted in Eq.15 may lead to incorrect $x^\ast$. For example, when the function is non-convex, maximizing its Jenson low bounder produce an estimate that is far away from $x^\ast$.

**Questions:**

1. Why is an i.ni.d noise model used to characterize the real-world noise?
2. Why do we want to adaptively update the parameters in reverse diffusion? Please provide some high-level explanation.
3. Please justify the accuracy of the approximation used in Eq. 15.
4. In Fig.3, self2self shows the best numerical performance. Please discuss the result.

**Limitations:**

The authors have discussed the limitations.

--------------------- After Rebuttal --------------------
The responses have addressed my questions. I have raised my score to 5.

---

> ### Author Rebuttal · Authors · 2024-08-05
>
> Thanks for your review of our paper. As there are some overlaps between Questions and Weaknesses, we will consolidate and answer them together.
>
> Q1: The paper does not well motivate their choice of i.ni.d noise model. It is not clear why such an i.ni.d noise model can properly characterize the statistics of the real-world noise considered in the paper. Why is an i.ni.d noise model used to characterize the real-world noise?
>
> Reply: As analyzed in Section 3.2 of the main paper, the real-world image noise is spatially correlated and signal-dependent. A structured and heteroscedastic Gaussian likelihood function can theoretically model real-world noise, but it is computationally expensive due to the large covariance matrix and also impractical due to the unknown noise variance. The reason we chose *i.ni.d.* Gaussian likelihood function $p(y_0|x_0, \phi_0)=\mathcal{N}(y_0;x_0, \phi_0^{-1})$ to model real-world noise is that it allows us to model the spatially variant feature (*ni* in *i.**ni**.d*. means not identical) of real-world noise and meanwhile frees the modeling of covariance (*i* in **i**.ni.d. means independent), as already explained in lines 147-149 of Section 3.3. Such a choice trades off the modeling precision for practical feasibility. Based on this *i.ni.d.* Gaussian model, we can dynamically estimate the noise precision posterior during the inverse diffusion process and then obtain an optimal $x^{*}_{t-1}$ that balances the prior and measurement. As shown in Figure 4 of the main paper, the estimated variance of *i.ni.d.* noise model well matches the noise level of noisy images and is *signal-dependent*, implying that the estimated *i.ni.d.* model is suitable for real-world noise.
>
> Another feasible choice is the *i.i.d.* Gaussian likelihood combined with our adaptive likelihood estimation (ALE) for the precision **scalar** (not the **vector** in *i.ni.d.* model). This model assumes the same noise variance across the spatial locations of real-world images. We compare this variant against our method and show the result in the following Table:
>
> |Datasets|CC|FMDD|
> |-|-|-|
> |*i.ni.d.*+ALE (ours)| **38.01/0.959**| **33.14/0.860**|
> |*i.i.d.*+ALE| 33.66/0.856|27.29/0.549|
>
> It is observed that adopting *i.ni.d.* noise model is necessary for real-world denoising and significantly outperforms the *i.i.d.* noise model
>
> Q2: The paper is not well-written, lacking proper motivation and explanation of the design of the proposed method.
>
> Reply: As replied to Q1, we would like to argue that the motivation of introducing the *i.ni.d.* Gaussian likelihood for real-world noise has been explained in Section 3.2 and Section 3.3 of the main paper, please see lines 137-144 and lines 146-149. This *i.ni.d.* noise model has also been checked in Figure 4(a) of Section 4.3, where the non-identical variances successfully capture the spatially varying noise level in noisy images.  Nevertheless, we noticed that such motivation was not mentioned in the Introduction section, which may confuse readers and impede the readability. We will modify our introduction in the revised version to include this motivation.
>
> Q3: Why do we want to adaptively update the parameters in reverse diffusion? Please provide some high-level explanation.
>
> Reply: The reason we estimate the precision $\phi_{t-1}$ during the inverse diffusion process is that specifying the accurate noise precision for each spatial location of $y_{t-1}, t=[0,\cdots T]$ is difficult and impractical, as indicated by lines 94-96, 150-152 of the main paper. Therefore, we assigned $\phi_{t-1}$ a prior and adaptively infer its posterior using variational Bayes. The updated precision successfully captures the spatially varying noise of noisy images, see Figure 4(a) in Section 4.3. Without the prior modeling and adaptive likelihood estimation, the denoising performance significantly degraded, as indicated in Table 3 of the main paper.
>
> Q4: The approximation adopted in Eq.15 may lead to incorrect x∗. For example, when the function is non-convex, maximizing its Jenson low bounder produce an estimate that is far away from x∗. Please justify the accuracy of the approximation used in Eq. 15.
>
> Reply: We argue that as $\log(\cdot)$ is a concave function and $\log p(y_{t-1}|x_{t-1})$ in Eq. (15) equals to $\log E_{g(\phi_t)}p(y_{t-1}|x_{t-1}, \phi_{t-1})$, applying Jensen's inequality **guarantees** to find its lower bound $E_{g(\phi_{t-1})}\log p(y_{t-1}|x_{t-1}, \phi_{t-1})$. Optimizing the lower bound generally produces satisfactory solutions, like in variational inference, VAE, and diffusion models.
>
> We can empirically check the approximation accuracy in Eq. (15) by approximating $p(y_{t-1}|x_{t-1})$ via Monte-Carlo integration $\frac{1}{M}\sum_{s=1}^Mp(y_{t-1}|x_{t-1}, \phi^s_{t-1}), \phi^s_{t-1} \sim g(\phi_{t-1})$. Then the solution $\bar{\pi}_ {t-1}y_ {t-1} + (1-\bar{\pi}_ {t-1}\mu_\theta(x_t, t))$ with $\bar{\pi}_ {t-1} = \frac{\sigma_{t}^2}{\sigma_{t}^2 + \frac{1}{M}\sum_{s=1}^M (\phi^s_ {t-1})^{-1}}$ approaches to the optimal $x^\dagger_{t-1}$, which can be considered as the ground-truth MAP estimate.
>
> We compare the denoising performance of using $x^\dagger_{t-1}$ against using $x^*_{t-1}$ in Eq. (15) in the following Table, which shows that the two methods perform similarly and hence indicates the approximation error is small.
>
> |Method|Using $x^\dagger_{t-1}$ (GT)| Using $x^*_{t-1}$ (in Eq. (15))|
> |-|-|-|
> |SIDD  Val|34.78/0.888|34.76/0.887|
> |CC|38.01/0.958| 38.01/0.959
>
> Q5: In Fig.3, self2self shows the best numerical performance. Please discuss the result.
>
> Reply: We note that Self2Self only performs well on real-world images with relatively light noise, e.g., PolyU and CC datasets, as shown in Table 3. And self2self happened to achieve the largest PSNR for that image of PolyU in Figure 3. However, its numerical performance on the whole PolyU underperforms ours. Moreover, Self2Self fails to handle heavy noise, as shown in Table 3 and Figure 5 of the paper.

---

> > ### Comment · Reviewer_tZQC · 2024-08-12
> > **Thanks for the response. Score raised.**
> >
> > I thank the authors for the thorough response to my comments. I found them very clear and helpful for me to better understand the motivation of the noise model and derivation of the proposed method. After a careful re-evaluation, I have raised my score from 3 to 5.

---

> > > ### Author Response · Authors · 2024-08-12
> > >
> > > Thank you for taking the time to read the rebuttal. We are glad that our response solved your questions and thank you for increasing your score.

---

### Official Review · Reviewer_eDuR · 2024-07-26

**Soundness:** 3
**Presentation:** 3
**Contribution:** 3
**Rating:** 6
**Confidence:** 3

**Summary:**

Overall, this paper presents a novel method in real-world image denoising by proposing a sophisticated method that combines adaptive likelihood estimation, MAP inference, and variational Bayes within the diffusion model framework.

**Strengths:**

1.	Utilizing variational Bayes to dynamically infer the precision posterior is innovative and provides a more accurate modeling of real-world noise.
2.	By exploring local priors in low-resolution diffusion models, the method can directly handle high-resolution noisy images without extensive patch-based or resize-based operations.
3.	The proposed method's effectiveness is demonstrated through extensive experiments on diverse real-world datasets, showcasing superior performance compared to other unsupervised denoising methods.

**Weaknesses:**

1.	The method relies on hyperparameters (e.g., prior precision, temperature parameter, kernel scale etc.), which might require careful tuning for optimal performance for different datasets. Authors criticize previous methods for being dependent on hyperparameters but do not overcome such limitation in this paper.
2.	The contribution of local Gaussian convolution might be overclaimed. As shown in  Table 3, the performance improvements brought by rectification are pretty limited, which look like experimental variations.

**Questions:**

1.	What is the NFE for DDRM used in this paper? What is the inference time for DDRM and your method respectively?
2.	In Algorithm 1, the threshold is set to 1e-6. Is performance sensitive to different values?

**Limitations:**

Authors discussed the limitation in the paper.

---

> ### Author Rebuttal · Authors · 2024-08-04
>
> Thanks for your recognition of our work.
>
> Q1: The method relies on hyperparameters (e.g., prior precision, temperature parameter, kernel scale etc.), which might require careful tuning for optimal performance for different datasets. Authors criticize previous methods for being dependent on hyperparameters but do not overcome such limitation in this paper.
>
> Reply: Our method does rely on some hyperparameters, but they are relatively robust for different choices. We ablate different $\beta$, temperature $\gamma$, and kernel scale $s$ on CC dataset, and report the corresponding performance in the following Tables:
>
> | $\beta$($\gamma=5,s=1$) | 5e-3 | 1e-2 | 1.5e-2 |
> | :--- |  :---: | :---: | :---:  |
> | PSNR/SSIM |38.03/0.957| 38.01/0.959 | 37.47/0.953|
>
> | $\gamma$($\beta=0.01,s=1$) | 1/4 | 1/5 | 1/10 |
> | :--- | :---: | :---: | :---: |
> | PSNR/SSIM| 37.90/0.957| 38.01/0.959| 37.80/0.953|
>
> | $s$($\gamma=5,\beta=0.01$)| 0.8 | 1.0 | 1.2 |
> | :--- | :---: | :---: | :---: |
> | PSNR/SSIM| 37.876/0.957| 38.01/0.959|38.10/0.959|
>
>
> It can be observed that the denoising performance resulting from different hyperparameters does not change significantly, indicating that they are quite insensitive to moderate changes. In addition, regarding compared methods like GDP and DR2, we carefully tuned their hyperparameters, but they still perform poorly in real-world denoising. In the modified version, we will highlight their inability to handle real-world noise rather than the reliance on hyperparameters.
>
> Q2: The contribution of local Gaussian convolution might be overclaimed. As shown in Table 3, the performance improvements brought by rectification are pretty limited, which look like experimental variations.
>
> Reply: We agree that the local Gaussian convolution introduced in Eq. (17) is a small technical improvement rather than one of the main contributions of our paper. Therefore, we did not include it into our main contributions, see lines 59-63 of the main paper. But we believe that the local Gaussian convolution is indeed helpful as it consistently enhances the denoising performance of all test datasets as shown in Table 3 regardless of the minor improvement on FMDD and PolyU datasets.
>
> Q3: What is the NFE for DDRM used in this paper? What is the inference time for DDRM and your method respectively?
>
> Reply: We used the DDIM sampling and 20 NFEs for DDRM, following its default setting. The inference time of DDRM for images with resolution $256\times 256$ is about 4s while our method costs about 230s. As discussed in the limitation section of the main paper, our method relies on the DDPM and large sampling steps for good performance. Speeding up the inference is our next research point.
>
> Q4: In Algorithm 1, the threshold is set to 1e-6. Is performance sensitive to different values?
>
> Reply: The denoising performance is insensitive to different thresholds as the alternate optimization in lines 5-8 of Algorithm 1 is guaranteed to converge, as indicated by [1]. In the following table, we ablate different thresholds on CC dataset, and results indicate that the threshold is robust.
>
> | Threshold      | 1e-5 | 1e-6 | 1e-7 | 1e-8 |
> | ----------- | ----------- | ----------- | ----------- | ----------- |
> | PSNR/SSIM     | 38.00/0.958 | 38.01/0.959       | 38.01/0.959      | 38.01/0.959 |
>
> [1] Bishop C M, Nasrabadi N M. Pattern recognition and machine learning[M]. New York: springer, 2006.

---

> > ### Comment · Reviewer_eDuR · 2024-08-13
> >
> > Thank you so much for the response. I have no further questions and will maintain my initial rating.

---

### Official Review · Reviewer_UCEq · 2024-07-29

**Soundness:** 2
**Presentation:** 4
**Contribution:** 2
**Rating:** 5
**Confidence:** 4

**Summary:**

The paper proposes a method to tackle real world noise using an adaptive likelihood estimation . For this the authors develop a technique to dynamically infer the precision posterior using variational bayes. The authors perform comprehensive evaluation on real world denoising datasets to show the effectiveness of the method.

**Strengths:**

1. The authors propose an method that could adaptively estimate the posterior distribution and make efficient use of off the shelf methods for real world denoising
2. The authors find that local priors from diffusion models pre-trained with LR image are more effective in real world denoising tasks.
3. The paper is well written and easy to follow. Derivations seems to be accurate.

**Weaknesses:**

1. The authors have utilized a gamma based hyperprior without reasoning the design choice or comparing with other possible prior distribution. In order to show the need for a gamma prior, further experimental analysis are required
2. The methods seems to be applicable only for real world denoising limiting its practical utility.
3. Analysis with key diffusion based restoration methods like [1,2] are missing
    [1] Diffusion Posterior Sampling for General Noisy Inverse Problems
    [2] Manifold Preserving Guided Diffusion
4. [2] achieves good results with just 50 steps of sampling.Could the authors provide an analysis of the variation of performance with different number of sampling steps.

**Questions:**

1. Can the authors provide an intuition of utilizing a gamma hyperprior over a gaussian prior like in [1,2].
[1] Diffusion Posterior Sampling for General Noisy Inverse Problems
[2] Manifold Preserving Guided Diffusion

2. Is the method generalizable to other linear inverse problems than denoising [1,2] and DDRM, are all generic methods providing a general formulation to all linear inverse problems.

3. Could the authors provide an analysis of the variation of performance with different number of sampling steps.

**Limitations:**

Yes, The authors have presented the limitations in a separate section.

---

> ### Author Rebuttal · Authors · 2024-08-04
>
> Thanks for your recognition of our work. As there are some overlaps between Questions and Weaknesses, we will consolidate and answer them together.
>
> Q1: The authors have utilized a gamma based hyperprior without reasoning the design choice or comparing with other possible prior distribution. In order to show the need for a gamma prior, further experimental analysis are required
>
> Reply: In our paper, we utilized the *i.ni.d.* Gaussian distribution to model real-world noise. And we were dedicated to dynamically estimating the noise precision $\phi_{t-1}$ at step $t$ by introducing the precision prior and conducting posterior inference. The reason we chose gamma prior for $\phi_{t-1}$ is that gamma distribution is the conjugate prior for the Gaussian likelihood function with unknown precision, and the corresponding precision posterior also follows a gamma. Such property is commonly used in variational Bayes and allows us to derive the closed-form expression for the precision posterior, see Eqs. (13, 14). As indicated by Table 3 in the main paper, without the prior modeling and variational Bayes, the denoising performance significantly degraded, highlighting the significance of introducing the gamma prior. Other informative priors are also feasible in principle, but there may be no closed-form posterior and the problem becomes intractable.
>
> We also experimented with a non-informative prior, Jeffery Prior. Particularly, Jeffery Prior for the precision of Gaussian results in closed-form posterior. If we choose $p(\phi_{t-1})\propto \frac{1}{\phi_{t-1}}$, then the precision posterior $g(\phi_{t-1})$ is a gamma distribution with shape $\hat{\alpha} _ {t-1}=\frac{1}{2 \gamma}$, and rate $\hat{\beta} _ {t-1}=\frac{(y_ {t-1} -\hat{\mu} _ {t-1} )^2 + \hat{\sigma}^2 _ {t-1}}{2 \gamma}$. Hence, we do not need to specify any extra parameters for $p(\phi_ {t-1})$ (i.e., $\alpha$ and $\beta$, refer to Eq. (14)). However, we found that such prior results in pure zero images and is useless, indicating the importance of utilizing informative priors, e.g., gamma.
>
> Q2: The methods seems to be applicable only for real world denoising limiting its practical utility. Is the method generalizable to other linear inverse problems than denoising [1,2] and DDRM, are all generic methods providing a general formulation to all linear inverse problems.
>
> Reply: The primary objective of this paper is to introduce diffusion priors for real-world image denoising, which is significant in photography, microscopy, and CT imaging. In addition to real-world denoising, Section 4.4 of the main paper also verified the effectiveness of our method on synthetic noise removal, including Poisson and Bernoulli noise. Moreover, our method is readily available for image restoration with pixel-wise degradation, including image demosaicing and inpainting.
>
> To adapt our method to these tasks, we define the forward process $y_0 = M \odot x_0$, where M is the degradation operator, and $\odot$ denotes element-wise multiplication. For image inpainting and demosaicing, $M$ is the binary mask with 0 values indicating missing pixels of $y_0$. We can incorporate $M$ into $p(y_{t-1}|x_{t-1},\phi_{t-1})$ in Eq. (15), which results in $\hat{\pi}_ {t-1 }=\frac{M\sigma^2_t}{M\sigma^2_t+1/\text{E}(\phi_ {t-1} )}$. We provide visual results of inpainting and demosaicing in Figure 1 of the 6958_rebuttal.pdf. In the following table, we also compare our method against DDRM on image demosaicing, and our method shows better results.
>
> |Demosaicing (CFA:RGGB)|Set14|CBSD68|
> |-|-|-|
> |DDRM|24.68/0.714|24.52/0.705|
> |Ours|**26.02/0.756**|**25.43/0.732**|
>
> When $M$ is not the pixel-wise degradation, our method needs more effort to adapt to.
>
> Q3: Analysis with key diffusion based restoration methods like [1,2] are missing.
>
> Reply: DPS [1] approximates the gradient of likelihood $\nabla_ {x_t} \log p_t(y_0|x_t)$ at step $t$ to $\nabla_ {x_t} \log p(y_0|\hat{x}_ 0)$, which acts as a hard data-consistency term that relates $y_0$ to $x_t$ in the generation process. If the forward model $p(y_0|x_0)$ is given, $x_t$ can be guided by $-s\nabla_{x_t}\log p(y_0|\hat{x}_0)$ with $s$ as the guidance level. MPGD [2] further proposed to first guide $\hat{x}_0$ based on $p(y_0|x_0)$, and then incorporated the modified $\hat{x}_0$ into the DDIM sampling. Although they perform well in general noisy inverse problems, they still rely on accurate noise models and are only verified on synthetic noise removal. The following experiment shows that DPS underperforms our method in real-world noise removal.
>
> |Datsets|SDDD Val|CC|PolyU|
> |-|-|-|-|
> |DPS|34.46/0.881|34.48/0.904|36.26/0.940|
> |Ours|**34.76/0.887**|**38.01/0.959** |**38.71/0.970**|
>
> Q4: Can the authors provide an intuition of utilizing a gamma hyperprior over a gaussian prior like in [1,2]
>
> Reply: The intuitive interpretation of our method is hijacking the unconditional $x_{t-1}$ at each step and forcing it to approach $y_{t-1}$. The modified $x^*_{t-1}$ is the convex combination of the mean of $x_{t-1}$ and $y_{t-1}$, with the coefficient $\hat{\pi}_ {t-1}$ defined by the relative magnitude between $\sigma^2_{t}$ and the inverse of gamma posterior $g(\phi_{t-1})$. We provide a visual illustration of our method in Figure 2 of the 6958_rebuttal.pdf.
>
> Q5: [2] achieves good results with just 50 steps of sampling.Could the authors provide an analysis of the variation of performance with different number of sampling steps.
>
> Reply: We ablate different sampling steps in the following:
>
> | Steps| $T=1000$|$T=500$|$T=250$|
> | -| - | - | - |
> | SIDD Val| **34.76/0.887**|33.54/0.838| 23.89/0.825|
> | CC| **38.01/0.959**|37.18/0.947|22.72/0.716|
>
> It is observed that $T=500$ degrades the performance, which is acceptable though. When $T=250$, the denoising fails.  As pointed out in the limitation Section, our method relies on DDPM and requires large sampling steps for good performance. Speeding up the inference is our key direction in the future.

---

> > ### Comment · Reviewer_UCEq · 2024-08-13
> >
> > Dear authors,
> >
> > I thank you for the rebuttal. After going through the rebuttal, I have decided to maintain my rating. The key factors impacting this score is the large number of diffusion steps needed for  good performance and the limited applicability to just denoising problems.

---

> > > ### Author Response · Authors · 2024-08-13
> > >
> > > Thanks so much for your reply. We acknowledge that the proposed method requires large sampling steps, and improving the sampling efficiency will be our next move. Regarding the application to other image restoration tasks, we show that it is straightforward to apply our method to pixel-wise degradation tasks, e.g., image inpainting and demosaicing, as replied to Q2. In the future, we would like to adapt our method to other non-pixel-wise degradations, e.g., image deblurring and super-resolution.

---

### Author Rebuttal · Authors · 2024-08-05

We thank all reviewers for their review of our paper. The response to each reviewer has been posted separately in the following. The 6958_rebuttal.pdf contains figures related to responses to Reviewers  UCEq and sawX.

---

### Author Response · Authors · 2024-08-11

Dear Reviewers,

We sincerely thank you for your precious time and efforts in reviewing our paper.

We would like to inquire whether our response has addressed your questions and concerns. We are more than happy to discuss with you further and provide additional materials.

Best regards,

The Authors

---

### Decision · Program_Chairs · 2024-09-25

**Decision:**

Accept (spotlight)

**Comment:**

This paper presents a novel method for real-world image denoising using diffusion priors combined with adaptive likelihood estimation and Maximum A Posteriori (MAP) inference. The method effectively integrates variational Bayes to dynamically infer the precision posterior, leading to a more accurate modeling of real-world noise (Reviewers eDuR, sawX, and Wzuv). The paper is well-written, with clear derivations and thorough experimental evaluation across multiple real-world datasets, showcasing superior performance compared to other unsupervised denoising methods (Reviewers UCEq, sawX, Wzuv).

After a careful consideration of the reviewers' comments, AC recommends accepting this paper. Despite some concerns regarding the justification of certain design choices and the computational cost, the paper presents a good contribution to the field of image denoising. The combination of diffusion priors with adaptive likelihood estimation is both novel and effective, addressing a critical problem in real-world imaging.